# Chronic Stress Blocks the Endometriosis Immune Response by Metabolic Reprogramming

**DOI:** 10.3390/ijms25010029

**Published:** 2023-12-19

**Authors:** Chong Lu, Jing Xu, Ke Li, Jing Wang, Yilin Dai, Yiqing Chen, Ranran Chai, Congjian Xu, Yu Kang

**Affiliations:** 1Gynecology Department, The Obstetrics and Gynecology Hospital of Fudan University, 419 Fangxie Road, Shanghai 200011, China; 2Shanghai Key Laboratory of Female Reproductive Endocrine-Related Diseases, Shanghai 200011, China

**Keywords:** chronic stress, endometriosis, immune response, metabolic reprogramming

## Abstract

Studies have shown that the occurrence and development of endometriosis are closely linked to long-term psychological stress. The specific contribution of chronic stress to the metabolic adaptations in patients with endometriosis is still unknown. Lesions were removed from ten endometriosis patients during an operation, and the participants were divided into two groups using a psychological questionnaire. An mRNA Human Gene Expression Microarray analysis was applied to compare the mRNA expression profiles between the chronic stress group and the control group. In addition, the reliability of the mRNA Human Gene Expression Microarray analysis was verified by using research on metabolites based on both the liquid chromatography (LC-MS/MS) technique and quantitative reverse transcription polymerase chain reaction (RT–PCR). A microarray analysis of significantly up-regulated, differentially expressed genes between the chronic stress and the control groups showed genes that were principally related to metabolism-related processes and immune-related processes, such as the immune response process, negative regulation of T cell proliferation, the leucine metabolic process, and the L-cysteine metabolic process (*p* < 0.05). LC-MS showed that the differential metabolites were primarily concerned with arginine and proline metabolism, D-glutamine and D-glutamate metabolism, aspartate metabolism, glycine, serine metabolism, and tyrosine metabolism (*p* < 0.05). The possibility of chronic stress blocks the endometriosis immune response through metabolic reprogramming. Chronic stress reduces the supply of energy substrates such as arginine and serine, down-regulates T immune cell activation, and affects the anti-tumor immune response, thereby promoting the migration and invasion of endometriosis lesions in patients with chronic stress.

## 1. Introduction

Endometriosis (EM) is a condition in which endometrial cells and stroma abnormally grow outside the uterus, affecting 10–15% of women of reproductive age [1]. The pathophysiology of endometriosis involves increased cell adhesion, neoangiogenesis, abnormal cell apoptosis, and genetic factors, but it remains poorly understood. Clinically, the condition manifests itself with dysmenorrhea, dyspareunia, non-cyclic pelvic pain, and decreased fertility, all factors that have a major impact on the quality of life of affected individuals [2,3,4]. Accordingly, women with endometriosis are more likely to live under chronic stress, including depression, anxiety, and low social support [5,6,7,8]. In particular, patients with deep-infiltrating pain arising from endometriosis may have a very high level of chronic stress; this, in many cases, can be relieved following surgical treatment [9].

Accumulating evidence has validated that chronic stress plays a crucial role in the progression of endometriosis [10,11,12]. However, the specific contribution of chronic stress to the development of endometriosis has not been fully delineated.

Chronic stress is a complex process that can activate the sympathetic nervous system (SNS) and the hypothalamus–pituitary–adrenal (HPA) axis [13,14]. It can cause a series of deleterious downstream effects on the patient’s body [15]. Both epinephrine (E) and norepinephrine (NE) are reported to be increased continuously in patients under chronic stress, and dopamine (DA) levels are decreased [16,17]. Chronic stress also mediates the GC receptor (GCR), which is part of the feedback system that regulates inflammatory and immune responses [18].

Appleyard and her colleagues demonstrated that prior exposure to swimming stress ahead of the surgical induction of endometriosis in rats caused a significant increase in both the size and number of lesions [19,20]. Guo and his colleagues claimed that chronic stress accelerates endometriosis development and is mediated by the activation of Adrenoceptor Beta 2 (ADRB2) and cyclic-AMP response binding protein (CREB) pathways, and that antagonism of ADRB2 signaling might be a new therapeutic avenue for the treatment of endometriosis [11,21]. Just as the relationship between chronic stress and cancer progression involves the SNS and HPA [15], a study showed that cancer patients reporting high levels of chronic stress had abnormal NE and cortisol levels [7]. However, the roles of chronic stress in endometriosis development are not completely understood.

Almost all living systems can be changed according to the microenvironment to regulate metabolism. Therefore, the cells need to monitor the change in the microenvironment at any time and alter their metabolic demands. However, different cells do not have the same metabolic adaptations. Normal cells mainly use glucose metabolism within mitochondria oxidative phosphorylation to produce adenosine triphosphate (ATP). However, tumor cells in the body include another metabolic mode, which is glycolysis. And under the condition of sufficient oxygen, tumor cells still use glycolysis, which is neither economical nor efficient compared with other processes, namely, the aerobic glycolysis process (Warburg effect) [22]. The metabolic condition of tumor cells affects not only itself, but at the same time can also affect the surrounding cells, such as innate and adaptive immune cells, endothelial cells, and tumor-associated fibroblasts in the immune system [23]. With the tumor cells’ growth, the immune cells in the microenvironment undergo metabolic reprogramming, which causes phenotypic changes [23]. The major scientific problem investigated in this study is how the tumor cell’s metabolic reprogramming process works to affect tumor immunity, which is why, in this study, we analyze metabolite differences and immune differences as well as how they relate to each other.

In our study, we performed an mRNA Human Gene Expression Microarray analysis to study the difference in the mRNA expression profiles between chronic stress and control endometriosis patients. In addition, using research on metabolites based on both the LC-MS/MS technique and quantitative RT–PCR, we verified the reliability of the mRNA Human Gene Expression Microarray analysis. Our aim was to discover the underlying mechanisms of chronic stress in the development of endometriosis and, perhaps, to devise potential biomarkers to assess the effect of acceleration of chronic stress on the invasion of endometriosis.

## 2. Results

### 2.1. Patient Characteristics

A total of 30 patients were recruited for this study, and they were categorized into a chronic stress group and a control group based on their psychological rating scales. All patients were diagnosed with endometriosis via pathological diagnosis in our study. The depression and anxiety of endometriosis patients were studied using the psychological questionnaires Patient Health Questionnaire-9 (PHQ-9) and Generalized Anxiety Disorder-7 (GAD-7). Age and stage, as mixed factors, were excluded (Figure 1a,b).

### 2.2. Differential Gene Expression in Endometriosis Patients with or without Chronic Stress

The mRNA Human Gene Expression Microarray (Agilent Sure Print G3 Human Gene Expression v3 (8 × 60 K, Design ID: 072363)) was used to profile mRNA expression. Five samples from each group were sent for mRNA testing. This demonstrated that there are a large number of differentially regulated genes between endometriosis patients with and without chronic stress. In the hierarchical cluster, endometriotic tissue from endometriosis patients with stress has a distinct gene expression profile from patients without stress. The dendrogram also shows that a large number of differentially expressed genes are up-regulated or down-regulated in endometriotic tissue from endometriosis patients with stress (Figure 2a,b).

In total, 34,757 expressed genes were detected in the endometriosis tissue. Of these genes, we identified 1381 mRNAs that were differentially expressed in the chronic stress group tissues versus control tissues. A total of 689 of these mRNAs were up-regulated and 692 were down-regulated in the chronic stress group tissues.

### 2.3. Gene Ontology (GO) Enrichment Analysis of Differentially Expressed Genes

To gain further insights into the biological functions of differentially expressed genes, we performed GO analyses by querying each differentially expressed gene identified in the chronic stress group and the control group. GO analysis of significantly up-regulated differentially expressed genes between the chronic stress group and control group revealed genes that were mainly enriched in immune-related processes and metabolism-related processes, such as the immune response process, the inflammatory response, the immune system process, regulation of the immune response, negative regulation of T cell proliferation, the leucine metabolic process, and the L-cysteine metabolic process (*p* < 0.05) (Figure 3). Down-regulated differentially expressed genes were mainly enriched in immune-related processes, while up-regulated differentially expressed genes were mainly involved in metabolism-related processes.

### 2.4. Pathway Analysis of Differentially Expressed Genes

We used the KEGG database to determine significant pathways involving the differentially expressed genes identified in this study. Our results showed that 14 pathways were significantly enriched for the identified DEGs. Moreover, pathway analysis showed that these differentially expressed genes were mainly involved in complement and coagulation cascades; cell adhesion molecules (CAMs); cytokine–cytokine receptor interaction; phagosomes; the NF-kappa B signaling pathway (NF-κB); cysteine and methionine metabolism; glycine, serine, and threonine metabolism; and D-Arginine and D-ornithine metabolism (Figure 4a,b).

### 2.5. Pathway–Act Network Analysis

To further understand the importance of pathway interactions and to screen key pathways for significant roles, we built a pathway–act network according to the direct or systemic interactions assigned between pathways in the KEGG database (Figure 5). As shown in Figure 6, some differentially expressed genes involved in key pathways in the chronic stress group and the control group were identified, including metabolic pathways such as glutathione metabolism; cysteine and methionine metabolism; and glycine, serine, and threonine metabolism, which were found to be up-regulated. Moreover, the FoxO signaling pathway, the Jak-STAT signaling pathway, and the NF-kappa B signaling pathway in the interaction network were also predicted to play important roles.

### 2.6. Differences in Metabolic Profiles and Pathways in Endometriosis Patients with or without Chronic Stress

All 30 samples were sent for metabolite testing. A total of 3629 known metabolites were detected and quantified in the endometriosis tissue. We identified the differential metabolites between endometriosis patients with and without chronic stress. A total of 235 out of 3629 metabolites were differential across these two groups (*p* < 0.05). To visualize the relationship between the 235 altered metabolites, hierarchical clustering was used to arrange the metabolites on the basis of their relative levels across samples (Figure 6). And pathway analysis showed that these differential metabolites were mainly involved in glutathione metabolism; D-glutamine and D-glutamate metabolism; arginine and proline metabolism; taurine and hypotaurine metabolism; alanine, aspartate, and glutamate metabolism; glycine, serine, and threonine metabolism; arachidonic acid metabolism; tyrosine metabolism; and purine metabolism (*p* < 0.05) (Figure 7). These showed almost the same changes as those of differential pathways, which was determined by using gene expression profile analysis for pathways of metabolic difference.

### 2.7. Quantitative Real-Time PCR Validation of Gene Expression Related to the Inflammatory Pathway and the Metabolic Pathway

To validate the expression profile results via real-time PCR quantification, we selected 10 mRNAs from 10 endometriosis samples (with 5 samples in each group) to be subjected to the expression profile analysis. The qRT-PCR results showed that the miRNA expressions of IL-1B, CXCR4, GABARAPL1, IL-7R, STAT4, CD14, CCR5, GATM, and GSTM2, as well as the mRNA expressions of CXCR4, GABARAPL1, IL-7R, STAT4, CD14, and CCR5, were significantly decreased (*p*-values for all <0.05) in the chronic stress group samples compared with the control group samples. And the mRNA expressions of GATM and GSTM2 were notably increased (*p*-values for all <0.05) in the chronic stress group samples compared with the control group samples (Figure 8). The qRT-PCR results were consistent with those obtained from the expression profile analysis, thus confirming the expression profile results.

## 3. Discussion

To investigate the underlying mechanisms of the promotional effect of chronic stress on endometriosis, we profiled the expression of mRNAs in endometriosis samples from 10 patients with and without chronic stress using Human Gene Expression Microarray analysis. We found that 1381 mRNAs were differentially expressed between the two groups. GO and pathway analyses were performed to predict the potential functions of the differentially expressed mRNAs, which shows that down-regulated differentially expressed genes were mainly enriched in immune-related processes, while up-regulated differentially expressed genes were mainly involved in metabolism-related processes. Moreover, we predicted the target genes of the differentially expressed mRNAs by constructing a pathway–act network. Based on our results, we predicted that metabolic pathways such as arginine, glycine, serine, and threonine metabolism would be found to be up-regulated. Additionally, the FoxO signaling pathway, the Jak-STAT signaling pathway, and the NF-kappa B signaling pathway in the interaction network were also predicted to play important roles. And metabolic profiles and qRT-PCR were applied to verify the analysis results.

Pain and infertility are the two major stressors that cause depression and anxiety in women with endometriosis [24,25], and these stressors are often largely uncontrollable and last a long time. Consequently, they activate SNS, which regulates the catecholamine–HPA axis, which in turn regulates glucocorticoid secretion [15]. As for endometriosis, glucocorticoid receptor expression is elevated and, thus, anti-glucocorticoid treatment has been advocated for [26,27]. However, the mechanisms of how the glucocorticoids demonstrated any effect on the development of endometriosis are still uncertain. The effects of other neuroendocrine substances mediated by chronic stress on the development of endometriosis are also unclear, although the effect on cancer development is much more well known [28,29].

The mRNA Human Gene Expression Microarray analysis indicated that, when comparing patients who were exposed to chronic stress with those who were not, differences in the immune and metabolic pathways were found in different genes. Tasuku and his colleagues reported that activation of NF-KB can enhance the expression of intercellular adhesion molecules (ICAM-1), MCP-1, and E-selectin; increase the endothelial permeability; and promote adhesion of the extracellular matrix (ECM), which causes an inflammatory response in endothelial cells. NF-κB promoted the overexpression of IL-8, which is conducive to angiogenesis in the ectopic endometrial matrix [30].

Nuclear factor-kappa B (NF-induced B) is a kind of nuclear transcription factor that widely exists in various cells of the body. It is closely related to immune and inflammatory reactions as well as the apoptosis, formation, and metastasis of tumors [31]. The activity of NF-κB on EM may regulate inflammatory response, adhesion, invasion, and angiogenesis. Wickiewicz et al. reported that the promoter of IL-6 has a binding site for NF-κB, and that NF-κB can induce IL-6 production and significantly increase IL-6 levels in the abdominal fluid of EM patients [32].

Li et al. showed that the combined DNA activity of NF-κB, p50, p65, VEGF, and COX-2 in the ectopic endometrial stromal cells was significantly increased, and the activated NF-κB pathway could up-regulate the expression of VEGF, which promotes angiogenesis and the degradation of the surrounding matrix, in addition to further stimulating the metastasis and regeneration of the ectopic endometrium [33]. In our study, chronic stress is indicated to up-regulate the expression of NFκB in the ectopic endometrium, which is conducive to the adhesion, implantation, and angiogenesis of the ectopic endometrium.

The Janus kinase/signal transducer is an activator of the transcription (JAK/STAT) pathway; it is a signal transduction pathway stimulated by cytokine and composed of three components: tyrosine kinase-related receptor, tyrosine kinase JAK, and transcription factor STAT [34]. The Janus kinase/signal transducer is involved in cell proliferation, differentiation, apoptosis, and immune regulation. When the JAK/STAT pathway is activated abnormally, cell proliferation and apoptosis-related genes such as cyclin D1, survivin, bcl-2, and bcl-xl are up-regulated, which promotes disease development [35]. Tu et al. found that STAT3 phosphorylation was significantly higher in the normal endometrium secretion phase than in the proliferative phase. In addition, STAT3 in the ectopic endometrium of EM patients was in an overactive state, and its phosphorylation level did not change with the menstrual cycle, suggesting that the abnormal activation of STAT3 in EM patients may be related to the abnormal ability of cells to decidualize differentiation [36]. STAT3 is involved in the balance regulation of immune escape and immune killing, and activated STAT3 can up-regulate the expression of immunosuppressive factors such as IL-10 and TGF-β and reduce the expression of immune activation factors such as IL-12, CD80, and CD86 [37]. Okamoto et al. applied the gene expression profile chip technology analysis, showing that STAT3 inhibitors can inhibit cell proliferation and VEGF and promote apoptosis, which indicates that STAT3 inhibitors are expected to be a candidate medicine for EM [38]. At the same time, our study found that this pathway was significantly inhibited in patients without chronic stress.

There are also a substantial number of studies on the effects of chronic activation of the stress response on the immune response associated with the development of endometriosis [24,39,40]. Research has shown that the abnormal dysfunction of natural killer cells, T lymphocytes, macrophages, mast cells, and NKT cells may lead to ectopic endometrial planting; moreover, abnormal expression of integrin ICAM-1, E-cadherin, and other adhesion molecules may be involved in the positioning and abnormal adhesion of Ems. Because of the abnormal immune mechanism, the activated immune cells release IL, TNF-α, VEGF, and a series of cytokines, which further aggravate the disorder of the immune mechanism [41,42,43,44]. And chronic stress tends to suppress the non-specific and specific components of the immune response, such as antigen presentation, cytotoxic T cell activity, NK cell activity proliferation, and production of inflammatory cytokines through mechanisms that involve adrenergic and glucocorticoid-mediated pathways [45]. Our results show that several mRNAs and pathways related to immune function, such as IL-1B, CXCR4, IL-7R, STAT4, CD14, and CCR5, as well as the FoxO signaling pathway, the Jak-STAT signaling pathway, and the NF-kappa B signaling pathway, have significant correlations with endometriosis.

Studies have shown that metabolism can affect the progress of endometriosis. TGF-1 and lactate levels of endometriosis patients are significantly higher than in normal females, and, in particular, the lesions of the disease are highly expressed in glycolytic-related genes, such as HIf1-α, Glut1, Pdk1, and Ldha, as well as the expression of glycolytic related genes; the glycolytic metabolism of the heterotopic endometrium was transferred to the warfarin effect [46,47]. Our study also found that chronic stress alters the microenvironment of endometrium cells and affects their implantation and invasion by affecting multiple metabolic pathways in the lesions of patients with endometriosis.

Tumor cells can select different metabolic ways to produce ATP and biomacromolecules for their own use according to the concentration and content of external nutrients such as glucose, glutamine, serine, arginine, and fatty acids [22]. When a lack of glucose or glutamine is present, cancer cells can activate oncogenes, such as cMyc, by regulating the expression of metabolic enzymes PHGDHP, SAT1, and PSPH in the serine synthesis pathway; by using the remaining glutamine or glucose support serine from synthetic approaches; and by maintaining REDOX steady-state support, such as tumor cell survival in nutritional stress conditions [48]. The metabolic patterns of tumor cells are complex and changeable, and they choose the best metabolic patterns to survive according to their environment. Interestingly, these metabolic patterns also exist in immune cells.

The immune system contains a variety of immune cells: the cells in the body when in a steady state or a static state, and cells in the body due to infection, inflammation, or other foreign materials that are activated and respond quickly [49]. Therefore, different metabolic patterns can influence the differentiation and function of different immune cells to affect the occurrence and development of tumors in the tumor microenvironment [50].

Tumor immune research shows that the production of lactic acid can affect the function of immune cells to promote the development of tumors. Early clinical studies have found that the degree of the patient’s tumor load in the serum lactate levels increased significantly. Further research has shown that lactic acid, instead of lactate, leads to CTL (cytotoxic T lymphocytes) lactic acid in the cells via acidification of the microenvironment, which cannot suppress CTL cell proliferation or the secretion of cytokines and cell toxicity [51,52,53]. Under the condition of the tumor microenvironment and tumor-similar macrophages, the high expression of VEGF and ARG1 by up-regulated HIF-1 α and lactic-acid-mediated HIF-1 α is stable under constant upstream oxygen material so as to activate VEGF and ARG1, which eventually results in tumor-associated macrophage (TAM) to M2 type polarization, and ARG1 secreted by TAM2 promotes tumor growth [54].

Amino acids and their metabolites from tumors also affect immune cells and their function. It is reported that the arginine that tumor cells absorb in the tumor microenvironment is mainly provided by tumor-associated bone marrow cells (macrophages, monocytes, myeloid cells, neutrophils, etc.) [55]. These immune cells help tumor cells to tolerate arginine’s lack of a microenvironment. And large data analyses in the latest research have shown that the activation of T cells will consume large amounts of arginine and add downstream products, and that exogenous glycine can increase the content of intracellular arginine and their downstream products by combining the transcription factor (BAZ1B, PSIP1, and TSN) binding protein model induced by metabolite glycolysis to OXPHOS transformation, promote the survival of T cells and the number of memory cells, and increase the antitumor immune response [56].

Studies have shown that metabolism can affect the progress of endometriosis, and TGF-1 and lactate levels of endometriosis patients are significantly higher than in normal females. In particular, the lesions of the disease are highly expressed to glycolytic-related genes such as HIF-1 α, Glut1, Pdk1, and Ldha. In addition to the expression of glycolytic-related genes, the glycolytic metabolism of the heterotopic endometrium was transferred to the warfarin effect [46].

Our study also found that chronic stress alters the microenvironment of endometrium cells. It also affects their implantation and invasion by affecting multiple metabolic pathways and immunological pathways in the lesions of patients with endometriosis.

In conclusion, our study demonstrates that chronic stress accelerates the development of endometriosis in EM patients. It also demonstrates that amino acids such as arginine and serine were more highly accumulated in the lesions of the chronic stress group compared with the control group. Meanwhile, the expressions of several immune-related pathways were down-regulated, which suggests that chronic stress may block the endometriosis immune response via metabolic reprogramming; the exact mechanism remains to be researched further. Given the current availability of related treatments such as stress management to blunt the promotional effect of stress on endometriosis development, it seems that chronic stress is a modifiable risk factor for endometriosis.

## 4. Materials and Methods

### 4.1. Patients

Patients diagnosed with endometriosis via intraoperative pathology were recruited from the Fudan University Affiliated Obstetrics and Gynecology Hospital. Psychological assessment scales, including the patient health questionnaire-9 (PHQ-9) and the generalized anxiety disorder-7 (GAD-7) scales, were applied to assess the psychological status of the patients (sup). Our assessment of the psychologic state of each patient was completed before the surgery, and all patients were informed of the purpose. The patients completed questionnaires under the guidance of professional instructors. We categorized patients with mild or higher levels (a score of 5 and above) of both depression and anxiety as the chronic stress group and those without either anxiety or depression as the control group. Tissue specimens were collected during the surgery, placed in sterile tubes, and immediately frozen at −80 °C for further analysis.

### 4.2. Gene Expression Profiling: RNA Extraction, Library Preparation, RNA-Sequencing, and Analyses of Differentially Expressed Genes

The Agilent Sure Print G3 Human Gene Expression v3 Microarray (8 * 60 K, Design ID: 072363) was used in this experiment, and data analysis of the 10 samples was completed.

The total RNA from the endometriosis samples was isolated using TRIzol reagent (Invitrogen, Carlsbad, CA, USA) on the basis of the manufacturer’s instructions. Total RNA was quantified using the NanoDrop ND-2000 (Thermo Scientific, New York, NY, USA), and the RNA integrity was assessed using the Agilent Bioanalyzer 2100 (Agilent Technologies, USA). The sample marking, microarray hybridization, and washing were carried out in accordance with the manufacturer’s protocols. Briefly, total RNA was transcribed into double-stranded cDNA, and cRNA was synthesized and labeled with Cyanine-3-CTP. Then, the labeled cRNAs were hybridized to the microarray. After washing, the microarrays were scanned using the Agilent Scanner G2505C (Agilent Technologies, Santa Clara, CA, USA).

Feature Extraction software (version 10.7.1.1, Agilent Technologies, Santa Clara, CA, USA) was applied to analyze the array images in order to obtain original data. Gene spring (version 13.1, Agilent Technologies, Santa Clara, CA, USA) was applied to complete the basic analysis with the original data. First, the quantile algorithm was used to normalize the original data. At least 100% of the values in any 1 of all conditions were tagged as “Detected” and selected for further data analysis. Then, *p* values were calculated via repeated changes and *t*-tests to identify differentially expressed genes. The thresholds of up-regulated and down-regulated genes had a fold change of ≥1.5 and *p* ≤ 0.05. GO analysis and pathway analysis were then used to determine the roles of these differentially expressed mRNAs. In the end, hierarchical clustering analysis was performed to show the expression patterns of the distinguishable genes among samples.

### 4.3. Metabolome Analyses: Tissue Metabolite Extraction and Metabolite Profiling Analysis

A total of 30 tissue samples of endometriosis (with 15 chronic stress patients and 15 control) were prepared in order to detect metabolites via LC-MS/MS analyses. LC-MS/MS analyses were applied using a UHPLC system (1290, Agilent Technologies, Santa Clara, CA, USA) with a UPLC BEH Amide column (1.7 μm 2.1 * 100 mm, Waters, Milford, MA, USA) coupled to Triple TOF 6600 (Q-TOF, AB Sciex). The mobile phase included 25 mM NH4OAc and 25 mM NH4OH in water (pH = 9.75) (A) and acetonitrile (B), according to elution gradient, as follows: 0 min, 5% A; 0.5 min, 5% A; 7 min, 35% A; 8 min, 60% A; 9 min, 60% A; 9.1 min, 5% A; and 12 min, 5% A, delivered at 0.3 mL/min^−1^. The injection volume was 2 μL. In LC/MS experiments, MS/MS spectra were obtained on the basis of information dependence using a triple TOF mass spectrometer. In this mode, the acquisition software (Analyst TF 1.7, AB Sciex, Framingham, MA, USA) continuously evaluates the full scan measurement MS data while collecting and triggering MS/MS spectral collection according to pre-selected criteria. In each cycle, 6 precursor ions with strength greater than 100 were selected and fragmented under impact energy (CE) of 35 V (15 MS/MS event, with an accumulation time of 50 msec for each production). ESI source conditions were set as follows: ion source gas 1 and 2 at 60 Psi, curtain gas at 30 Psi, source temperature of 550 °C, and ion spray voltage floating (ISVF) values of 5500 V and −4500 V in the positive and negative mode, respectively.

### 4.4. Quantitative RT–PCR

Total RNA was extracted from frozen tissues and reverse-transcribed into cDNA using the PrimerScript RT Kit (Takara, Shiga, Japan) in accordance with the manufacturer’s protocols. Quantitative real-time PCR (qRT-PCR) was applied with SYBR Premix Ex Taq (Takara, Japan) on a 7500 RT-PCR system (Applied Biosystems, Foster City, CA, USA). Each sample was analyzed in triplicate. The mRNA expression levels were normalized to the 18 s level and measured via the 2^−ΔΔCt^ method.

### 4.5. Statistics

The experimental data are shown as the mean ± standard deviation. One-way ANOVA was adopted for comparing more than two groups of means, and the shortest effective range was used for comparison between the two groups. The SPSS 20.0 software (IBM Statistics Inc., Chicago, IL, USA) was applied for statistical analysis. Graphical representations made performed with GraphPad Prism 6.0 software (San Diego, CA, USA), and Student’s *t*-tests were used for the intergroup comparison. The differences between the groups were statistically significant at values of *p* < 0.05.

## Figures and Tables

**Figure 1 ijms-25-00029-f001:**
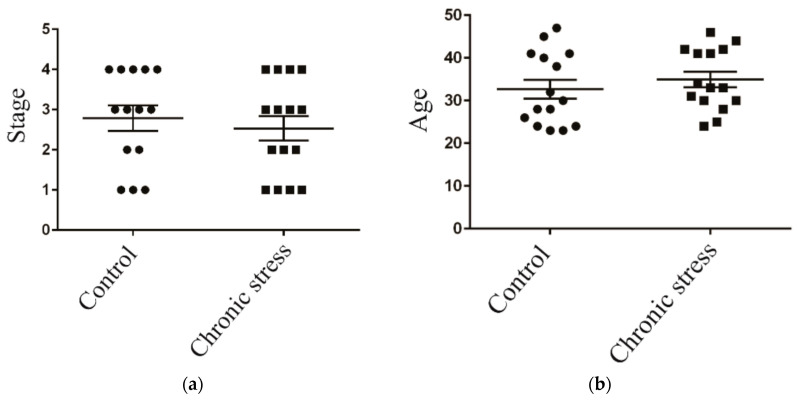
Patient characteristics. (**a**) Scatter plot analysis of age in control group and chronic stress group of metabolome analysis patients; (**b**) scatter plot analysis of stage in control group and chronic stress group of metabolome analysis patients.

**Figure 2 ijms-25-00029-f002:**
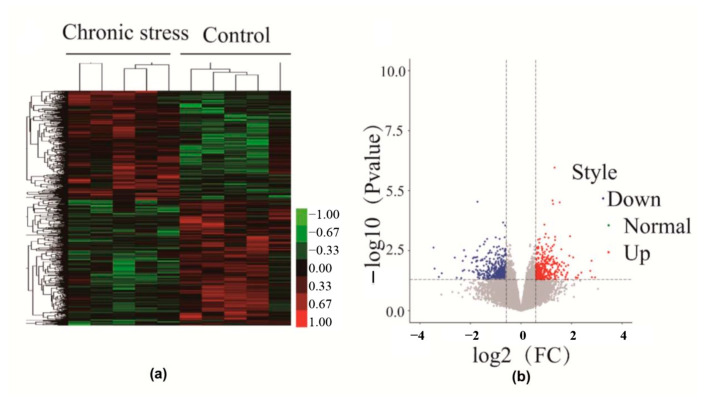
(**a**) Heat map of the mRNA profiles in endometriosis tissues of the chronic stress group versus paired control endometriosis tissues. Red: up-regulated genes; green: down-regulated genes. The expression of mRNA is hierarchically clustered on the y-axis, and the tissue samples are hierarchically clustered on the x-axis (fold change: 2; *p* < 0.05). Codes marked with numbers and letters were obtained from five chronic stress endometriosis tissues and five paired control endometriosis tissues, respectively. (**b**) The volcano plot numbers in the overlapping regions indicate that genes similarly regulated according to the selected genes had a ≥2-fold change (*p* < 0.05) for chronic stress endometriosis tissue compared with the control.

**Figure 3 ijms-25-00029-f003:**
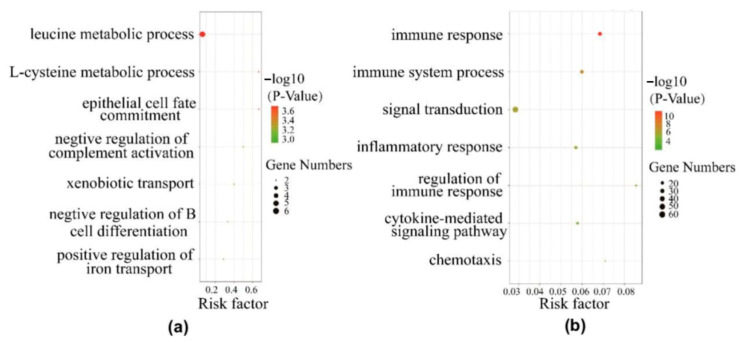
Gene ontology (GO) of differentially expressed (DE) genes. The vertical axis represents the GO category and the horizontal axis represents the −Log10 (*p*-value) of the significant GO terms. Red bubbles indicate corrected *p* < 0.05. (**a**) Top 7 significant GO terms (biological processes) associated with the identified up-regulated DE genes. (**b**) Top 7 significant GO terms (biological processes) associated with the identified down-regulated DE genes.

**Figure 4 ijms-25-00029-f004:**
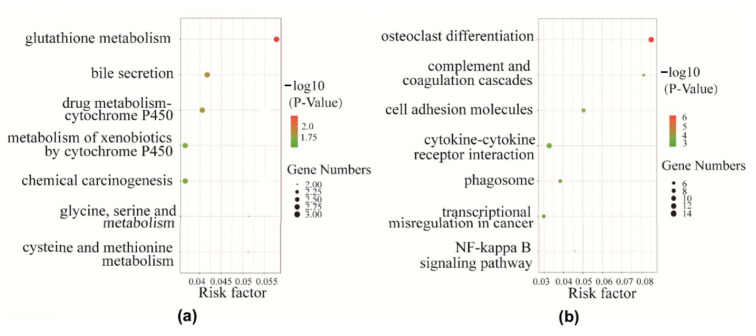
Pathway analysis of differentially expressed genes. The vertical axis represents the GO category and the horizontal axis represents the −Log10 (*p*-value) of the significant pathway terms. Red bars indicate corrected *p* < 0.05. (**a**) Top 7 significant pathways associated with the identified down-regulated differentially expressed genes. (**b**) Top 7 significant pathways associated with the identified up-regulated differentially expressed genes.

**Figure 5 ijms-25-00029-f005:**
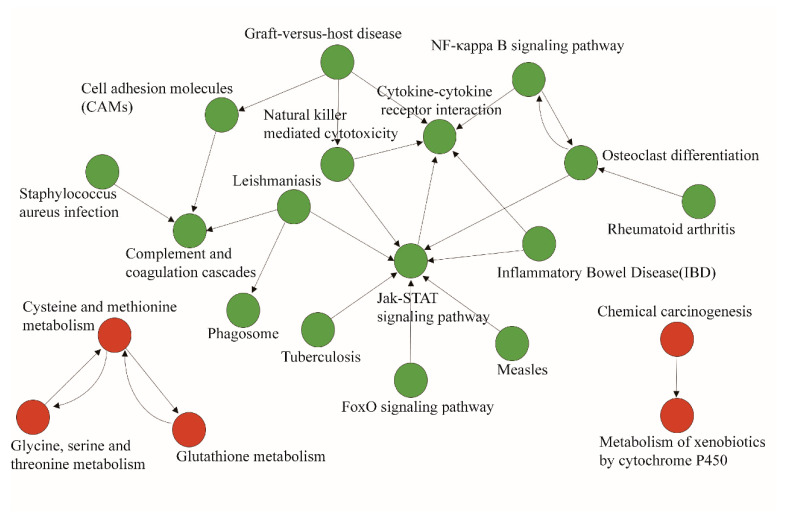
Pathway–act network analysis. The pathway–act network was built according to interactions with pathways identified in the KEGG database. Cycle nodes represent pathways, and arrows between two nodes represent interaction targets between pathways. Red nodes represent up-regulated pathways and green nodes represent down-regulated pathways.

**Figure 6 ijms-25-00029-f006:**
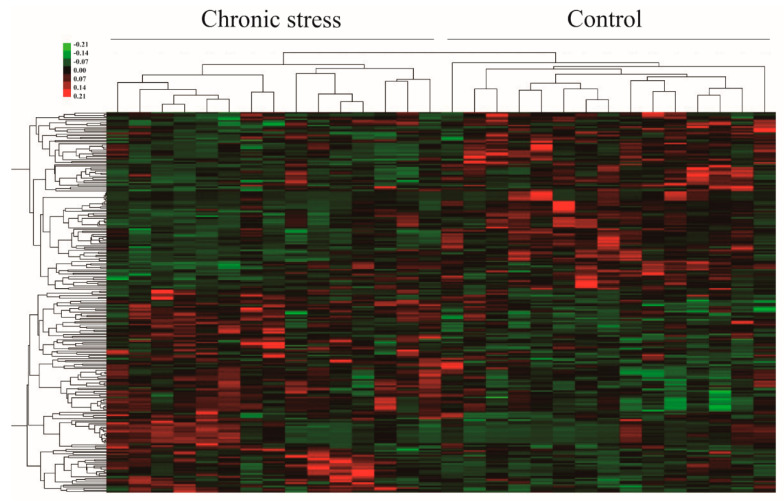
Heat map of the metabolic profiles in endometriosis tissues of the chronic stress group versus the paired control group endometriosis tissues. Red: up-regulated metabolites; green: down-regulated metabolites. The levels of metabolites are hierarchically clustered on the y-axis, and the tissue samples are hierarchically clustered on the x-axis (fold change: 2; *p* < 0.05). Codes marked with numbers and letters were obtained from 15 chronic stress endometriosis tissues and 15 paired control endometriosis tissues, respectively.

**Figure 7 ijms-25-00029-f007:**
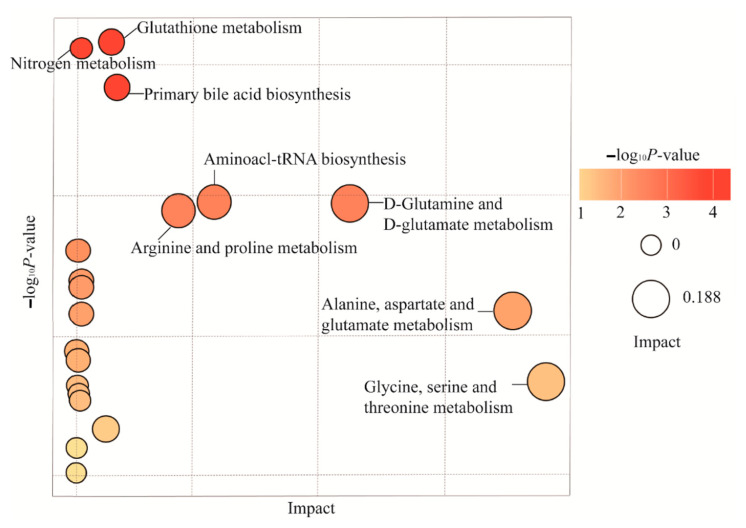
Bubble charts of the differential metabolic pathways in endometriosis tissues of the chronic stress group versus the paired control group.

**Figure 8 ijms-25-00029-f008:**
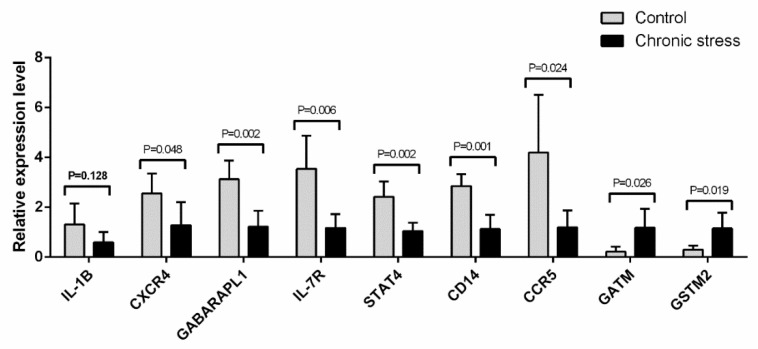
Validation of expression profile data using quantitative real-time PCR. Ten mRNAs (IL-1B, CXCR4, GABARAPL1, IL-7R, STAT4, CD14, CCR5, GATM, and GSTM2) were selected and analyzed via qRT-PCR to validate their expression levels. The relative expression level of the target mRNA was normalized. The data displayed in the histograms are expressed as means ± standard deviation (SD).

## Data Availability

Due to the nature of the research, due to ethical supporting data is not available.

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
