# Peer review of "Chronic Stress Blocks the Endometriosis Immune Response by Metabolic Reprogramming"

_ijms, 2023, doi:10.3390/ijms25010029_

Round 1

Reviewer 1 Report

Comments and Suggestions for Authors

High quality paper. Interesting data of endometriosis  from metabolic point of view.  According to authors, chronic stress reduces the supply of energy substrate such as arginine and serine, and down-regulate the T immune cells activation and promote invasion of endometriosis lesions. It can be important in introduction new treatment modalities in patients with endometriosis.

Author Response

Thank you very much for your comments.

Reviewer 2 Report

Comments and Suggestions for Authors

''Chronic Stress Blocks the Endometriosis Immune Response by Metabolic Reprogramming''

The manuscript presented for review consists of 14 pages with 55 references (most of them are not up-to-date). 8 figures are included.  The manuscript is divided into 4 sections (Introduction, Results, Discussion, Material and Methods). The work fits the journal scope. Moderate English changes are required (grammar, sentence structures). I would recommend to check the article by a native speaker. The aim is defined. Keywords are adequate and refer to the whole context.

Introduction:

-The abbreviation EM - should be explained in the brackets.

-Line 38: ''it can be obviously relieved after the surgical therapy'' - not always.

-Line 64 - ATP -  should be explained in the brackets.

Material and methods:

-''10 patients with/without chronic stress'' -  small groups of patients

-Line 393-394: ''30 tissue samples of endometriosis (with 15 chronic stress patients and 15 control) were prepared'' - How many patients were in control and study group? 10 or 15?

Statistical tests are correct. 

Comments on the Quality of English Language

Moderate English changes are required (grammar, sentence structures). I would recommend to check the article by a native speaker.

Author Response

Thank you for your constructive advice. We have employed professional Language Editing Services provided by MDPI to enhance the English expression of this article.

Reviewer 3 Report

Comments and Suggestions for Authors

This study evaluated the impact of chronic stress on the development of endometriosis. The study group should be more properly defined. It is not clear what were the inclusion and exclusion criteria. How was endometriosis diagnosed (laparoscopy, US...)? Is it possible to draw any definite conclusions with only 10 included patients. What are the possible clinical implications of this study?

Author Response

Thank you very much for your valuable advice. The patients in this study were diagnosed by intraoperative pathology. We used the PHQ-9 scale to assess the depression level of the patients,  a score of 5 and above was considered mild depression, and the GAD-7 scale to assess the anxiety level of the patients, and a score of 5 and above was considered mild anxiety. We divided the patients with depression and anxiety into the experimental group and the patients with negative scores on both scales into the control group. The number of patients was the main limitation of this study, but the conclusions drawn from our study suggests that we should pay attention to the psychological state of patients with endometriosis.

Reviewer 4 Report

Comments and Suggestions for Authors

The object of this investigation is of interest since it deals with an area that has not really been investigated.

In spite of this, before the content of the manuscript can be critically evaluated the text must be reviewed by a native English-speaking person, because of imprecisions and even errors.

I wanted to make suggestions on how to improve the text, but in the end I was forced to give-up in view of the extension of the changes required.

To show what should be done, I have made suggestions below on how to improve the first few paragraphs of the Introduction. 

Endometriosis is a condition in which endometrial cells and stroma abnormally grow outside the uterus; it can affect 10-15% of women of reproductive age [1]. However, The pathophysiology of endometriosis involves increased cell adhesion, neoangiogenesis, abnormal cell apoptosis, and genetic factors, but it remains poorly understood. Clinically, the condition manifests itself with dysmenorrhea, dyspareunia, non-cyclic pelvic pain, decreased fertility, all factors that have a major impact on the quality of life of affected individuals [2-4]. Accordingly, women with endometriosis are more likely to live under chronic stress including depression, anxiety and low social support [5-8]. Especially patients with deep infiltrating pain arose by endometriosis may have a very high level of chronic stress; this, in many cases can be obviously relieved following surgical treatment [9].

There is accumulating evidence has validated that chronic stress plays a crucial role in the progression of endometriosis [10-12]. However, its specific contribution of 40 chronic stress to the development of endometriosis has not been fully delineated.

Chronic stress is a complex process which can activate the sympathetic nervous system (SNS) and hypothalamus-pituitary-adrenal (HPA) axis [13, 14]. It can cause a series of deleterious downstream effects on the patients’ body [15].

The following are general comments:

1.       When an abbreviation is utilised for the first time in both the Abstract and in theTtext, it should be fully explained.

2.       The first sentence in the Abstract (“Studies have shown that the occurrence and the development of endometriosis are closely linked to long-term psychological stress”) is confusing. The sentence is affirmative and is contradicted by the one that immediately immediately follows. Also the text (lines 33-36) indicates that subhects with endometriosis may suffer from chronic stress and its consequences. Therefore, todate no proof seems to exist of 'metabolic effects'.

In other words, the effects of chronic stress are the object of the present investigation and not a pathogenetic mechanism for the development of the condition.

3.       Personally, I do not like at all presenting Material and Methods after the Discussion. The reader should be informed of the nature of the clinical material and of the methods utilised before reading the results.

4.       In terms of “Clinical Material” utilized for this investigation, this was composed of “patients diagnosed with endometriosis”.

The first missing specificiation is “how many” patients were recruited.

From Figure 1 one would assume that two groups of 5 subjects each were utilised for the study of the “mRNA Human Gene Expression Microarray” [(a) and (b)].

Also from Figure 1 it can be deduced that for the “Metabolome analysis”, 14/15 subjects served as controls and 15 were considered as with chronic stress.

First of all, a statistician should evaluate the results and determine whether these numbers are such to lead to meaningful conclusions.

5.        From my point of view data do not mean much without specific information on the type of endometriosis of the various subjects (superficial peritoneal; ovarian; deep endometriosis). Also, the stage according to one of the widely accepted classification systems, is important to establish the validity of the study,

Given the fundamental nature of the information missing, it does not seem useful at this stage to go into minor issues.

Comments on the Quality of English Language

see my comments above

Author Response

  1. When an abbreviation is utilised for the first time in both the Abstract and in theTtext, it should be fully explained.

Response: Thank you for your valuable suggestion. We have conducted a detailed check for abbreviations that appear in this article to ensure that they are labeled in detail the first time they appear.

  1. The first sentence in the Abstract (“Studies have shown that the occurrence and the development of endometriosis are closely linked to long-term psychological stress”) is confusing. The sentence is affirmative and is contradicted by the one that immediately immediately follows. Also, the text (lines 33-36) indicates that subjects with endometriosis may suffer from chronic stress and its consequences. Therefore, to date, no proof seems to exist of 'metabolic effects'.

In other words, the effects of chronic stress are the object of the present investigation and not a pathogenetic mechanism for the development of the condition.

Response: Thank you for your valuable suggestion. The objective of our study was to investigate the specific contribution of chronic stress on the metabolic adaptations in patients with endometriosis and search for the biomarker associated with chronic stress in endometriosis.

3. Personally, I do not like at all presenting Material and Methods after the Discussion. The reader should be informed of the nature of the clinical material and of the methods utilised before reading the results.

Response: Thank you for your valuable suggestion. The manuscript was written according to the submission format required by the journal.

  1. In terms of “Clinical Material” utilized for this investigation, this was composed of “patients diagnosed with endometriosis”.

The first missing specification is “how many” patients were recruited.

From Figure 1 one would assume that two groups of 5 subjects each were utilised for the study of the “mRNA Human Gene Expression Microarray” [(a) and (b)].

Also from Figure 1 it can be deduced that for the “Metabolome analysis”, 14/15 subjects served as controls and 15 were considered as with chronic stress.

First of all, a statistician should evaluate the results and determine whether these numbers are sufficient to lead to meaningful conclusions.

Response: Thank you very much for your insightful comment. We recruited a total of 30 patients in this study, 15 patients each in the chronic stress group and the control group. Initially, we recruited 10 patients for RNAseq. At the time of follow-up metabolomics testing, we found that the sample size was insufficient and expanded to 15 patients in each group. 20 newly recruited patients did not undergo RNAseq testing due to financial reasons.

  1. From my point of view, data do not mean much without specific information on the type of endometriosis of the various subjects (superficial peritoneal; ovarian; deep endometriosis). Also, the stage according to one of the widely accepted classification systems, is important to establish the validity of the study,

Given the fundamental nature of the information missing, it does not seem useful at this stage to go into minor issues.

Response: Thank you for your constructive advice. Due to the limited sample size, we are unable to specify different types of endometriosis according to your suggestion. We have subsequently conducted clinical studies to further explore the impact of chronic stress on the metabolism of endometriosis, and these results are the only conclusions we have drawn by now.

Round 2

Reviewer 4 Report

Comments and Suggestions for Authors

See comments for the Editors

Comments on the Quality of English Language

None

Author Response

“Chronic Stress Blocks the Endometriosis Immune Response by Metabolic Reprogramming''

The manuscript presented for review consists of 14 pages with 55 references (most of them are not up-to-date). 8 figures are included.  The manuscript is divided into 4 sections (Introduction, Results, Discussion, Material and Methods). The work fits the journal scope. Moderate English changes are required (grammar, sentence structures). I would recommend to check the article by a native speaker. The aim is defined. Keywords are adequate and refer to the whole context.

Response: Thank you very much for your constructive advice. We have employed professional Language Editing Services provided by MDPI to enhance the English expression of this article.

Introduction:

-The abbreviation EM - should be explained in the brackets.

Response: Thank you very much for your constructive advice. In response to your suggestion, we have made the appropriate changes in the text, and all abbreviations in the text have been labeled when they first appear.

-Line 38: ''it can be obviously relieved after the surgical therapy'' - not always.

Response: Thank you very much for your valuable advice.The article lacks precision in expressing the meaning here, and in the revised article we have adjusted the expression accordingly.The point we are trying to make here is that some of our patients are in a better physical condition after the surgical treatment compared to the pre-surgical period, which in turn reduces their psychological stress.

-Line 64 - ATP -  should be explained in the brackets.

Response: Thank you very much for your constructive advice. In response to your suggestion, we have made the appropriate changes in the text, and all abbreviations in the text have been labeled when they first appear.

Material and methods:

-''10 patients with/without chronic stress'' -  small groups of patients

Response:

Thank you very much for your insightful comment.The number of patients was the main limitation in this study, but the conclusions drawn from our study can suggest that we should pay attention to the psychological state of patients with endometriosis.

-Line 393-394: ''30 tissue samples of endometriosis (with 15 chronic stress patients and 15 control) were prepared'' - How many patients were in control and study group? 10 or 15?

Response: Thank you very much for your insightful comment. We recruited a total of 30 patients in this study, 15 patients each in the chronic stress group and the control group. Initially we recruited 10 patients for RNAseq. At the time of follow-up metabolomics testing, we found that the sample size was insufficient and expanded to 15 patients in each group. 20 newly recruited patients did not undergo RNAseq testing due to financial reasons.

Statistical tests are correct. 

Response:Thank you very much for your insightful comment. 
